# Update of Robotic Surgery in Benign Gynecological Pathology: Systematic Review

**DOI:** 10.3390/medicina58040552

**Published:** 2022-04-17

**Authors:** Vito Andrea Capozzi, Elisa Scarpelli, Giulia Armano, Luciano Monfardini, Angela Celardo, Gaetano Maria Munno, Nicola Fortunato, Primo Vagnetti, Maria Teresa Schettino, Giulia Grassini, Domenico Labriola, Carla Loreto, Marco Torella, Stefano Cianci

**Affiliations:** 1Department of Medicine and Surgery, University Hospital of Parma, 43125 Parma, Italy; elisascarpelli13@gmail.com (E.S.); giulia.armano@gmail.com (G.A.); luciano.monfardini@gmail.com (L.M.); 2Department of Women, Child and General and Specialized Surgery, University of Campania “Luigi Vanvitelli”, 80138 Naples, Italy; angelacelardo@gmail.com (A.C.); gmm9401@gmail.com (G.M.M.); nicola.fortunato@libero.it (N.F.); primo.vagnetti@unicampania.it (P.V.); mariateresa.sche@gmail.com (M.T.S.); giulia.grassini9@gmail.com (G.G.); domenico.labriola@unicampania.it (D.L.); carla.loreto@unicampania.it (C.L.); marcotorella@iol.it (M.T.); 3Department of Gynecologic Oncology and Minimally-Invasive Gynecologic Surgery, Università degli Studi di Messina, Policlinico G. Martino, 98124 Messina, Italy; stefanoc85@hotmail.it

**Keywords:** robotic surgery, gynecology, myomectomy, hysterectomy, endometriosis, pelvic organ prolapse, minimally invasive surgery

## Abstract

*Background and Objectives**:* Since the Food and Drug Administration’s (FDA) approval in 2005, the application of robotic surgery (RS) in gynecology has been adopted all over the world. This study aimed to provide an update on RS in benign gynecological pathology by reporting the scientific recommendations and high-value scientific literature available to date. *Materials and Methods:* A systematic review of the literature was performed. Prospective randomized clinical trials (RCT) and large retrospective trials were included in the present review. *Results:* Twenty-two studies were considered eligible for the review: eight studies regarding robotic myomectomy, five studies on robotic hysterectomy, five studies about RS in endometriosis treatment, and four studies on robotic pelvic organ prolapse (POP) treatment. Overall, 12 RCT and 10 retrospective studies were included in the analysis. In total 269,728 patients were enrolled, 1721 in the myomectomy group, 265,100 in the hysterectomy group, 1527 in the endometriosis surgical treatment group, and 1380 patients received treatment for POP. *Conclusions:* Currently, a minimally invasive approach is suggested in benign gynecological pathologies. According to the available evidence, RS has comparable clinical outcomes compared to laparoscopy (LPS). RS allowed a growing number of patients to gain access to MIS and benefit from a minimally invasive treatment, due to a flattened learning curve and enhanced dexterity and visualization.

## 1. Introduction

In the last decades, minimally invasive surgery (MIS) was widespread both in benign and malignant pathologies [1,2]. Furthermore, since the Food and Drug Administration (FDA) approval in 2005, the application of robotic surgery (RS) in gynecology was adopted all over the world [3]. MIS is associated with a minor length of hospital stay, less blood loss, a reduction in postoperative pain, and superior long-term quality of life compared to the open approach [4]. Furthermore, the MIS approach is also encouraged by Enhanced Recovery After Surgery (ERAS) recommendations as a tool to improve fast recovery after surgery [5]. However, laparoscopy (LPS) and RS require a fair number of procedures for one to become confident with the surgical gestures, with a slow learning curve. LPS is characterized by two-dimensional visualization, a limited range of motions, difficulty with hand-eye coordination, and enhanced physiologic tremors [6]. Therefore, the introduction of RS provided the same LPS advantages with additional improvements. Moreover, RS with the 3D visualization, wristed instrumentation, and improved ergonomics can facilitate the surgical gestures of inexperienced surgeons [7]. However, due to the emerging technology and specific equipment, RS has higher costs and longer operative times compared to open and LPS approaches. In light of these data, the RS application presents known advantages for patients and surgeons but not always strong scientific evidence to support its use in clinical practice.

The present study aimed to provide an update on RS in benign gynecological pathology by reporting the scientific recommendations and the high-value scientific literature available to date. For this purpose, only randomized clinical trials (RCT) and large retrospective cohort studies are discussed in the present review.

## 2. Materials and Methods

A systematic review of the literature was performed in double-blind by two authors (VAC and ES). The analysis was conducted from September 2021 to January 2022. A third author (SC) checked the selected articles. Research on Pubmed, Web of Science, and Scopus was carried out using the following keywords: “robotic surgery” and “gynecology”, “robotic surgery” and “myomectomy”, “robotic surgery” and “hysterectomy”, “robotic surgery” and “endometriosis”, “robotic surgery” and “pelvic organ prolapse”, “robotic surgery” and “benign gynecological disease”. 

The agreement about potential relevance was reached by consensus of the researchers and according to PRISMA statement guidelines [8]. After the first selection, the authors evaluated the full-text copies of selected papers and separately extracted relevant data regarding study characteristics and outcomes. All bibliographies were analyzed to evaluate additional eligible studies. Only RCT and retrospective cohort trials were included in the present review in order to synthesize the relevant evidence about the current role of RS. Studies considered not in line with the purpose of the study, prospective non-randomized trials, case reports, analysis with a small number of patients (<20 cases), redundant studies, and articles not in the English language were excluded. Since no RCTs comparing robotic myomectomy to other surgical techniques have been published yet, only retrospective cohort studies were included in this case. 

## 3. Results

The electronic database search provided a total of 2130 studies published between 2005 and 2021. Of these, 258 duplicates, 781 case reports, 63 studies not in the English language, and 1009 works not fitting the review scope were excluded from the analysis. The study selection flowchart is shown in Figure 1.

Twenty-two studies were considered eligible for the study, eight studies regarding robotic myomectomy (Table 1), five studies on robotic hysterectomy (Table 2), five studies about RS in endometriosis treatment (Table 3), and four studies on robotic pelvic organ prolapse (POP) treatment (Table 4).

Overall, 12 RCT and 10 retrospective studies were included in the analysis.

The total of patients enrolled was 269,728, 1721 in the myomectomy group, 265,100 in the hysterectomy group, 1527 in the endometriosis surgical treatment group, and 1380 patients who received treatment for POP.

To better illustrate the results of the research and describe scientific evidence about different gynecological procedures, the main findings are reported in chapters: robotic myomectomy, robotic hysterectomy, robotic endometriosis eradication, and robotic pelvic organ prolapse treatment.

### 3.1. Robotic Myomectomy

Leiomyoma is the most common benign gynecologic tumor diagnosed in women during reproductive age. The true incidence in the general population is unknown because fibromas are often asymptomatic. However, almost 60% of women in reproductive age have fibroids [31]. The most common clinical presentation is abnormal uterine bleeding, bulk symptoms, and infertility. Fibroids can be classified depending on their uterine localizations according to the International Federation of Gynecology and Obstetrics (FIGO) [32]. Myomectomy is a safe treatment in symptomatic patients who desire to preserve their fertility. The fertility preserving surgical approach includes abdominal myomectomy (AM), laparoscopic myomectomy (LM), and robotic-assisted myomectomy (RAM). The appropriate surgical treatment should be individualized depending on myoma dimensions, number, localization, and surgeon skills.

In a large prospective randomized trial published in 2000 comparing LM and AM, LPS was associated with a minor length of hospital stay, less blood loss, smaller scars, faster recovery, and a non-inferiority pregnancy rate [33]. However, LPS is also characterized by some limitations. In the absence of a wide range of motion and limited visualization as in the case of a large uterus, laparoscopic dissection may be challenging. Besides, experts’ opinions suggest that LPS is contraindicated for fibroids greater than 10–12 cm and in the presence of more than three lesions requiring multiple uterine incisions.

RM has gained wide acceptance because robotic endowrist instruments offer better maneuverability and facilitated sutures. Moreover, RS is comparable to LPS in terms of enhanced recovery, perioperative outcomes, and cosmetic results. Limitations may derive from the lack of haptic feedback, in particular in controlling strength in suturing, in case of need for closure of cavity defect after myomectomy, and the individuation and location of small myomas. Furthermore, the removal of large myomas could be laborious due to the reduction of the surgical field visibility. To date, no randomized clinical trials comparing RM to open or laparoscopic approaches are available in the literature. However, retrospective non-inferiority trials support RM feasibility. In 2007, Advincula et al. published a retrospective case-matched study including 58 patients with symptomatic leiomyomas undergoing AM or RAM. The results showed higher postoperative complications, greater blood loss, and longer hospital stays in the AM group. Nevertheless, higher costs and longer operative times were reported in the RAM group [9]. In a retrospective analysis of 81 LM and RAM cases, Bedient et al. reported comparable short-term outcomes for both approaches, while long-term outcomes were not assessed [10].

Along with these results, Nezhat et al., in a retrospective case-matched study of 50 patients (35 LM and 15 RAM), reported longer operative times in the RS group and comparable post-operative short-term outcomes when compared to LM. Furthermore, the same authors reported that the main RAM advantage was the flattened learning curve that could allow less experienced endoscopic surgeons to perform MIS [11].

In 2012, with a large retrospective trial (115 LMs and 174 RAMs), Gargiulo et al. reported that LM and RAM have comparable short-term surgical outcomes [12]. RAM had longer operative times and larger estimated blood loss; however, during LM, a higher rate of the barbed suture were performed (67.9% vs. 5%), with significant impact on suturing time and blood loss. Subsequently, Barakat et al. published a large retrospective study on 575 myomectomies, comparing AM, LA, and RAM. The authors found that RAM was associated with decreased blood loss and less length of hospital stay compared with traditional LPS and AM. Furthermore, RM and LM shared comparable advantages compared to open surgery in terms of perioperative outcomes. However, myoma diameters were significantly higher in the robotic and open surgery arms compared to the laparoscopic group [13]. In line with these authors, Gobern et al. reported shorter hospital stays and decreased blood loss in the MIS group in a retrospective analysis of 308 procedures (169 AM, 73 LM, and 66 RAM) [14].

In a recent large retrospective trial conducted by Özbaşlı et al., the authors reported the safety and feasibility of a robotic-assisted approach in patients with large uterine size and myomas. Moreover, RAM patients experienced significantly reduced post-operative pain compared to AM and LM patients [16].

Long-term surgical outcomes were investigated in a retrospective study conducted by Flyckt et al. analyzing 133 myomectomies (80 AMs, 28 LM, and 25 RAM). After a median follow-up of eight years, women wishing for pregnancy showed a 55% pregnancy rate without a statistically significant difference in the three groups. Moreover, the bleeding symptom control was similar regardless of the surgical approach used [15]. Furthermore, no cases of uterine rupture were reported in the MIS groups [34]. In line with these authors, in a recent retrospective case series, Goldberg et al. reported a 70% pregnancy rate in 123 patients undergoing RAM [35].

In conclusion, in the absence of randomized prospective trials, several noninferiority studies are now available to indicate that RAM is as effective and safe as conventional LM. Moreover, the value of RS could offer a minimally invasive approach to patients that otherwise would be treated with open surgery. The limitations are related to the higher costs and longer operating time.

### 3.2. Robotic Hysterectomy

Hysterectomy is one of the most performed surgical procedures worldwide. In 90% of cases, benign pathologies are the main indication for the surgical procedure [36]. Surgical approaches to benign hysterectomy include laparotomy, LPS, vaginal and robotic techniques [37].

Over time, both open and vaginal approaches are decreasing in popularity, while the widespread adoption of robotic-assisted hysterectomy gave access to a larger number of patients to minimally invasive techniques, even in cases of severe obesity [38].

First, in 2009 and subsequently in 2021, the American College of Obstetricians and Gynecologists (ACOG) recommended the MIS approach as the gold standard for hysterectomy. Furthermore, among minimally invasive techniques, the vaginal route should be the primary choice whenever feasible [39].

Nevertheless, concern for malignancy, large uterine size, a fixed uterus, or the lack of confidence of the surgeon may preclude the vaginal approach. When the vaginal route is not indicated or feasible, LPS is mentioned as the preferred alternative to open surgery [40]. Advantages of laparoscopic hysterectomy (LH) over open abdominal hysterectomy (AH) include decreased postoperative pain, shorter hospital stay, and quicker return to daily activities [41]. However, the steep learning curve, counter-intuitive hand movement, as well as limited instruments movement and two-dimensional visualization are limitations of the technique [42].

On the other hand, robotic hysterectomy (RH) requires a lower level of technical skill with more intuitive surgical gestures compared to LH [43]. As a consequence, RH gained great popularity thanks to the easy adoption of the technique, even in the absence of strong evidence supporting RH over LH [44,45]. Despite lacking a strict indication for use of robotic-assisted hysterectomy, the RH may represent a suitable minimally invasive option in less optimal candidates for LPS. In particular, RS may offer a favorable alternative in severely obese patients [46].

In a population-based retrospective study conducted by Wright et al. on 264,758 women undergoing hysterectomy for benign disease, the authors found that LH and RH share comparable postoperative outcomes, although RS was associated with higher costs [17]. RH advantages and disadvantages were also assessed in randomized clinical trials. In a blinded, prospective randomized controlled trial conducted by Paraiso et al., 53 patients were randomized to LH (n = 27) and RH (n = 26). The authors reported a low complication rate for both approaches without statistically significant differences between the two groups. No intraoperative lesions or need for transfusions was registered. Furthermore, RH was associated with longer operative time, good postoperative pain control, and a fast return to daily activities [18]. In line with these results, Sarlos et al., in an RCT enrolling 95 patients who underwent LH or RH reported higher operating times in the robotic group and similar surgical and postoperative outcomes. Furthermore, patients enrolled in the RH arm reported a higher level of short term postoperative quality of life [19]. Lonnerfors et al. in an RCT with 122 patients (61 LH vs. 61 RH), also reported better short-term outcomes and a lower rate of postoperative complications in RH compared to the LH group. Concerning operative times, there were no differences between LH and RH. This may suggest that, where RS is well implemented, operating room time is not affected [20]. In agreement with this observation, Deimling et al. found no significant difference in operating time between LH and RH within the 144 cases analyzed. The mean operative time in the RH group was 73.9 min and 74.9 min in the LH group. The Authors concluded that RS when performed by experienced surgeons is not inferior to LPS in terms of operative time [21].

In summary, to date, there are no clear indications for RH over other minimally invasive techniques. At present, the main indications include patient obesity, uterine size, and surgeons’ expertise. For benign pathologies, RS appears non-inferior to LPS in the hands of expert surgeons, but with increased costs. The main advantage provided by RH adoption is a greater number of patients who gained access to a minimally invasive approach.

### 3.3. Robotic Endometriosis Treatment

Endometriosis is a chronic inflammatory condition that affects women during reproductive age. Endometriosis is associated with pelvic pain and infertility, but the severity of symptoms is not predictive of the stage of the disease. Endometriosis eradication is one of the most complex laparoscopic surgeries due to the distortion of the normal anatomy, adhesions, and hypomobility of the pelvic organs [47].

Surgical treatment depends on symptoms, response to medications, and women’s fertility status. Currently, although MIS is the approach of choice, no indication as to which MIS approach to prefer is present in the literature [48]. LPS is accepted as the preferred technique because of comparable outcomes to open surgery with the known advantages of MIS [49]. To date, scientific evidence about RS in endometriosis cases is limited. Many studies report that RS in endometriosis is a feasible and safe option [24,25,26]. However, most of these studies are retrospective in nature or with a limited number of cases reported. On the other hand, RS uses are reported in complex cases of deep infiltrating endometriosis with urinary and bowel involvement.

The LAROSE trial [22] is a prospective randomized clinical trial comparing LPS to RS in terms of operative times and perioperative outcomes in endometriotic disease. In the 53 patients enrolled (38 LPS vs. 35 RS), RS was shown to be non-inferior to LPS for both aspects. Even after adjustment for the stage of disease, operative times and quality of life after a six-month follow-up were similar. Further evidence comes from retrospective clinical trials [23,24,25,26] comparing robotic and laparoscopic approaches.

These retrospective studies showed minor operative times for LPS and superimposable complication rates compared to RS. However, due to their retrospective nature, comparison between LPS and RS is limited by the lack of randomization and the heterogeneity of stage of disease between the two approaches. Furthermore, surgeon experience and the need for other specialists in advanced stages should also be investigated.

In conclusion, MIS is the gold standard for endometriosis surgical treatment. Currently, both robotic and LPS are acceptable techniques for endometriosis surgical treatment. RS offers enhanced visualization and higher dexterity that can overcome some LPS limitations. Furthermore, according to the current evidence, RS could be the best option in complex cases with deep infiltrating endometriosis.

### 3.4. Robotic Pelvic Organ Prolapse Treatment

Pelvic organ prolapse (POP) is a common cause of morbidity in women with a remarkable impact on quality of life [50]. Surgery can offer a wide range of options to restore pelvic anatomy and function. The surgical approach depends on the surgeon’s experience, patient’s performance status, age, and patient comorbidity. The gold standard surgical treatment for grade 2–4 vaginal vault prolapse is sacrocolpopexy. Sacrocolpopexy superiority compared to other techniques, such as sacrospinous vaginal apex suspension, has been confirmed in a randomized clinical trial [51]. Furthermore, open abdominal sacrocolpopexy (ASC) is associated with optimal long-term outcomes.

However, the advent of MIS and its known advantages compared to open surgery made its application in POP surgery an issue of interest. In a randomized clinical trial by Freeman in 2013, non-inferiority of LPS vs. ASC in terms of perioperative outcomes and anatomic restoration according to the Pelvic Organ Prolapse Quantification System (POP-Q) were equivalent [52]. In addition, the robotic approach offers better visualization during dissection and easier suturing compared to LPS. As a consequence, RS may represent a feasible option for providing greater access to patients and surgeons to minimally invasive techniques due to a flatter learning curve [53]. Robotic and laparoscopic sacrocolpopexy has been compared in randomized clinical trials. Paraiso et al. enrolled 78 patients with 2–4 stage POP, 38 in the laparoscopic group, and 40 in the robotic group. Robotic sacrocolpopexy (RSC) was associated with longer operative time, increased postoperative pain, higher costs, and no benefits in terms of the anatomic and functional success of the technique after a one-year follow-up compared to laparoscopic sacrocolpopexy (LSC) [27].

Anger et al. randomized 78 patients with symptomatic POP to LSC and RSC. The primary outcome was to evaluate costs over the six weeks after surgery. Secondary outcomes were perioperative complications, postoperative pain, and clinical long-term outcomes after six months of follow-up. The results showed higher costs, longer operative time, and increased postoperative pain for the robotic approach with overlapping long-term outcomes compared to LSC. The author hypothesized that the lack of tactile feedback may hinder the surgeon’s control of pressure exerted on ports with a slight temporary increase in postoperative pain [28]. Furthermore, a high success rate for minimally invasive sacrocolpopexy was confirmed in an ancillary analysis after a one-year follow-up without differences between the two groups [54]. Illiano et al. in 2019 published a non-inferiority RCT comparing RSC to LSC for POP repair in patients with symptomatic POP-Q stage III-IV. Both arms showed excellent results with a 100% cure rate of apical compartment defect. RSC also showed a higher restoration rate of the anterior and posterior compartment compared to LSC but without statistical significance [29].

In a large retrospective trial published by Nosti et al. on 1124 patients (589 ASC vs. 273 LSC vs. 262 RSC), the open approach was associated with a higher rate of intraoperative and postoperative complications compared to minimally invasive sacrocolpopexy (MISC). MISC was associated with less blood loss, minor length of hospital stay, and longer operative time compared to ASC, especially in the robotic group. Furthermore, RSC patients experienced a minor rate of postoperative complications compared to LSC [30].

According to the available evidence, RSC may be considered a non-inferior alternative compared to LSC. The advantage provided by a flatter learning curve in the robotic approach may have value for surgeons with no experience in LPS. On the other hand, RS is associated with higher costs and longer operative times compared to LSC.

## 4. Conclusions

Currently, a minimally invasive approach is suggested in benign gynecological pathologies. According to the available evidence, RS has comparable clinical outcomes compared to LPS, but at the expense of higher costs and longer operating times. On the other hand, the introduction of RS has allowed a growing number of patients to gain access to MIS and benefit from a minimally invasive treatment due to a flattened learning curve and enhanced dexterity and visualization.

## Figures and Tables

**Figure 1 medicina-58-00552-f001:**
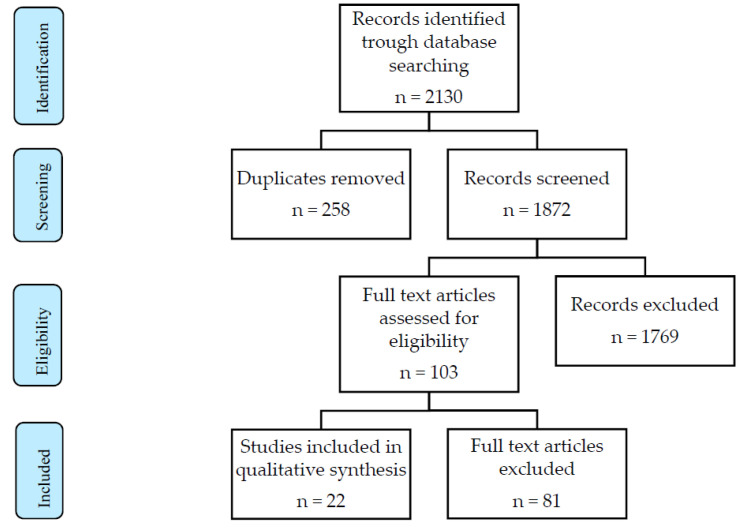
PRISMA flow diagram.

**Table 1 medicina-58-00552-t001:** Robotic Myomectomy studies.

Author, Year	Design of the Study	Surgical Approach	Sample Size	Main Findings	Short Term Outcomes	Long Term Outcomes
Advincula, 2007 [9]	Retrospective case-matched study	AM RAM	58 29 29	A robotic approach is associated with higher costs compared to laparotomy. Decreased estimated blood loss, complication rates, and length of stay with the robotic approach may prove to have a significant benefit.	Operative times: RAM > AM Estimated blood loss: RAM < AM Hospitalization time: RAM < AM Complications rate: AM > RAM (n 12 vs. 3)	NA
Bedient, 2009 [10]	Retrospective study	LM RAM	81 41 40	Short-term surgical outcomes were similar after robotic and laparoscopic myomectomy; long-term outcomes were not assessed.	Operative times: no significant differences Estimated blood loss: no significant differences Hospitalization time: no significant differences Complications rate: no significant differences	NA
Nezhat, 2009 [11]	Retrospective case matched study	LM RAM	50 35 15	RAM has a shorter learning curve, and does not add any additional morbidity to the LM. However, RAM shows no clinical advantage compared to LM. It may be useful during the learning period for non-experienced endoscopic surgeons	Operative times: RAM > LM Estimated blood loss: no significant difference Hospitalization time: no significant difference Complications rate: no major complications in the two groups	Pregnancy rate: no significant difference
Gargiulo, 2012 [12]	Retrospective cohort study	LM RAM	289 115 174	RAM and LM have similar operative outcomes. Operative time and intraoperative estimated blood loss were significantly greater in the robot-assisted laparoscopic myomectomy group. Use of barbed suture in the laparoscopic myomectomy group may account for these differences.	Operative times: RAM > LM Estimated blood loss: RAM > LM Hospitalization time: RAM > L Complications rate: no significant difference	NA
Barakat, 2011 [13]	Retrospective study	AM LM RAM	575 393 93 89	RAM is associated with decreased blood loss and length of hospital stay compared with LM and AM. Robotic technology could improve the utilization of the laparoscopic approach for the surgical management of symptomatic myomas.	Operative times: RAM > LM and AM Estimated blood loss: RAM < LM and AM Hospitalization time: AM > RAM and LM Complications rate: no major complications in RAM group, 1 wound dehiscence in LM, and 1 bowel injury in the LM	NA
Gobern, 2013 [14]	Retrospective study	AM LM RAM	308 169 73 66	LM and RAM demonstrated shorter hospital stays, less blood loss, and fewer transfusions than abdominal myomectomies. Robotic myomectomy offers a minimally invasive alternative for management of symptomatic myoma in a community hospital setting.	Operative time: RAM > LM and AM Blood loss: RAM and LM < AM Hospitalization time: RAM and LM < AM Postoperative pain: NA Complications rate: no differences	NA
Flyckt, 2016 [15]	Retrospective cohort study	AM LM RAM	133 80 28 25	There is no significant difference in long-term bleeding or fertility outcomes in robotic-assisted, laparoscopic, or abdominal myomectomy.	NA	Pregnancy rate: 60% with no differences between groups Uterine rupture: no cases Quality of life: no significant differences
Özbaşlı, 2021 [16]	Retrospective study	AM LM RAM	227 73 88 66	LM or RM may be a good choice for women of reproductive age because of short hospitalization duration, less blood transfusion and less postoperative pain. RAM appeared to be advantageous for patients with large myomas, on the other hand RM is more expensive and has longer operative times.	Operative time: RAM > LM and AM Blood loss: RM > LM and AM Hospitalization time: no differences Postoperative pain: RAM < LM and AM Complications rate: no significant differences.	NA

AM: abdominal myomectomy, LM: laparoscopic myomectomy, RAM: robotic assisted myomectomy, NA: not assessed.

**Table 2 medicina-58-00552-t002:** Robotic Hysterectomy studies.

Author, Year	Design of the Study	Surgical Approach	Sample Size	Main Findings	Short Term Outcomes	Long Term Outcomes
Wright, 2013 [17]	Retrospective cohort study	AH VH LH RH	264758 123288 54912 75761 10797	Between 2007 and 2010, the use of RH increased substantially. RH and LH had similar morbidity profiles, but the use of robotic technology resulted in more costs.	Hospitalization time: LH > RH Complications rate: 5.3% LH, 5.5% RH (no significant)	NA
Paraiso, 2013 [18]	RCT	LH RH	53 27 26	LH and RH are safe approaches to hysterectomy. RH requires a significantly longer operative time.	Operative time: RH > LH Blood loss: comparable Hospitalization time: comparable Postoperative pain: comparable Complications rate: no significant differences, no major complications.	Quality of life at six months: no significant difference
Sarlos, 2012 [19]	RCT	LH RH	95	RH and LH compare well in most surgical aspects, but the robotic procedure is associated with longer operating times. Postoperative quality of-life index was better; however, longterm, there was no difference.	Operative time: RH > LH Blood loss: no significant difference Hospitalization time: Postoperative pain: Complications rate: no significant difference.	Long term quality of life: no difference
Lonnerfors, 2015 [20]	RCT	MIS (LH and VH) RH	122 61 61	A similar hospital cost can be attained for laparoscopy and robotics when the robot is a preexisting investment. Robotic-assisted hysterectomy is not advantageous for treating benign conditions when a vaginal approach is feasible in a high proportion of patients.	Operative time: comparable Blood loss: RH < LH Hospitalization time: NA Postoperative pain: NA Complications rate: RH < LH and VH	NA
Deimling, 2017 [21]	RCT	LH RH	144 72 72	When performed by a surgeon experienced in both techniques, the operative time for RH was non-inferior to that achieved with LH.	Operative time: comparable Blood loss: comparable Hospitalization time: NA Postoperative pain: NA Complications rate: one ureter transection in RH group. No differences in postoperative complications	NA

AH: abdominal hysterectomy, LH: laparoscopic hysterectomy, VH: vaginal hysterectomy, RH: robotic hysterectomy, RCT: randomized controlled trial, MIS: minimally invasive surgery, NA: not assessed.

**Table 3 medicina-58-00552-t003:** Robotic endometriosis treatment studies.

Author, Year	Design of the Study	Surgical Approach	Sample Size	Main Findings	Short Term Outcomes	Long Term Outcomes
Soto, 2017 [22]	RCT	LPS RS	73 38 35	Laparoscopy and robotic surgery for the treatment of endometriosis have comparable perioperative outcomes, even after adjustment for stage of disease, and significant improvement in quality of life after intervention.	Operative time: comparable Blood loss: comparable Complications rate: comparable Quality of life at six weeks: comparable	Quality of life at six months
Dubeshter, 2013 [23]	Retrospective study	LPS RS	423 292 131	The results show a minor length of operative times for LPS, and comparable outcomes regarding complications and perioperative outcomes for both groups.	Operative time: comparable Blood loss: comparable Complications rate: comparable	NA
Magrina, 2015 [24]	Retrospective study	LPS RS	493 162 331	RS is associated with longer operating time. Operating time is an independent and significant factor for postoperative complications and hospital stay.	Operative time: RS > LS Blood loss: depending on operative time Hospitalization time: depending on operative time Complications rate: depending on operative time	NA
Nezhat, 2013 [25]	Retrospective study	LPS RS	118 86 32	Despite a higher operating room time, RS appears to be a safe minimally invasive approach for advanced stage endometriosis treatment, with all other perioperative outcomes, including intraoperative and postoperative complications, comparable with those in patients undergoing LPS.	Operative time: RS > LPS Blood loss: comparable Complications rate: comparable	NA
Nezhat, 2015 [26]	Retrospective study	LPS RS	420 273 147	LPS and RS are excellent methods for treatment of advanced stages of endometriosis. However, use of the robotic platform may increase operative time and might also be associated with a longer hospital stay.	Operative time: RS > LPS Blood loss: comparable Hospitalization time: RS > LPS Complications rate: comparable	NA

LPS: laparoscopy, RS: robotic surgery, NA: not assessed.

**Table 4 medicina-58-00552-t004:** Robotic pelvic organ prolapse treatment studies.

Author, Year	Design of the Study	Surgical Approach	Sample Size	Main Findings	Short Term Outcomes	Long Term Outcomes
Paraiso, 2011 [27]	RCT	LSC RSC	78 38 40	Robotic-assisted sacrocolpopexy results in longer operating time and increased pain and cost compared with the conventional laparoscopic approach.	Operative time: RSC > LSC Blood loss: comparable Postoperative pain: RSC > LSC Complications rate: comparable	one year functional outcomes and vaginal support: comparable
Anger, 2014 [28]	RCT	LSC RSC	78 38 40	Costs of robotic sacrocolpopexy are higher than laparoscopic, while short-term outcomes and complications are similar. Primary cost differences resulted from robot maintenance and purchase costs.	Operative time: RSC > LSC Postoperative pain: RS > LPS Complications rate: comparable	six months POP outcome: comparable
Illiano, 2019 [29]	RCT	LSC RSC	100 51 49	RSC provides outcomes as good as those of LSC with 100% anatomical correction of the apical compartment. RSC can be considered a good alternative in the treatment of symptomatic, stage III or IV, POP.	Operative time: RSC > LSC Blood loss: comparable Hospitalization time: comparable Complications rate: comparable	Urinary, anorectal symtpoms and sexual funtion improved in both groups without significant difference.
Nosti, 2014 [30]	Retrospective study	ASC LSC RSC	1124 589 273 262	ASC is associated with a higher rate of perioperative and postoperative complications compared to MISC. The MISC group had shorter length of hospitalization, less blood loss, and longer operative times. Within the MISC group, RSC was associated with fewer complications compared to LSC. There was no difference in anatomic failure with any sacrocolpopexy approach	Operative time: RSC > LSC Blood loss: ASC > RSC and LSC Hospitalization time: ASC > RSC and LSC Complications rate: RSC < LSC and ASC	No significant difference in the rate of anatomical failure between the ASC and MISC groups

POP: pelvic organ prolapse, ASC: abdominal socrocolpopexy, LSC: laparoscopic sacrocolpexy, RSC: robotic sacrocolpopexy, MISC: minimally invasive sacrocolpopexy, NA: not assessed.

## Data Availability

Not applicable.

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
