# Peer review of "Update of Robotic Surgery in Benign Gynecological Pathology: Systematic Review"

_medicina, 2022, doi:10.3390/medicina58040552_

Round 1
Reviewer 1 Report
I read with great interest the manuscript, which falls within the aim of this Journal. In my honest opinion, the topic is interesting enough to attract the readers’ attention. Nevertheless, authors should clarify some points and improve the discussion, as suggested below.
Authors should consider the following recommendations:
- Manuscript should be further revised in order to correct some typos and improve style.
- I would suggest to discuss, at least briefly, novel pieces of evidence about robotic surgery for the management of ovarian cancer and surgery for female-to-male transition (PMID: 27040423; PMID: 31795787)
Reviewer 2 Report
Thank you for the opportunity to comment on the work: “Update of robotic surgery in benign gynecological pathology”. Due to the popularization of the robotic-assisted laparoscopic approach to benign gynecological disease, the topic is clinically useful and relevant.
The Authors intended to provide a systematic review of the literature including prospective randomized clinical trials (RCT) and large retrospective trials. Finally, 19 studies were included: 5 studies on robotic-assisted myomectomy, 5 studies on robotic-assisted hysterectomy, 5 studies about robotic approach to endometriosis, and 4 studies on robotic pelvic organ prolapse.
Unfortunately, the submitted work has several shortcomings that should be improved upon careful revision.
1. Title: Please, identify the report as a systematic review (First item on the PRISMA checklist!)
2. Abstract: Please, include the abstract into the manuscript (the abstract is currently available only online)
3. Introduction and Discussion: The Authors do not mention important systematic reviews and meta-analyses with the same focus. It would be necessary to explain what is the novelty of their submitted work?
It is surprising why, e.g., a systematic review and meta-analysis of studies comparing robotic assisted myomectomy with other approaches - published in 2006 - included 15 articles, and the present work, including search until January 2022, only 5 articles. The same applies to hysterectomy etc. Please, consider following works with similar focus and corresponding methodology:
Marchand G et al. Systematic review and meta-analysis of all randomized controlled trials comparing gynecologic laparoscopic procedures with and without robotic assistance. Eur J Obstet Gynecol Reprod Biol. 2021, PMID: 34418694.
Iavazzo C et al. Robotic assisted vs laparoscopic and/or open myomectomy: systematic review and meta-analysis of the clinical evidence. Arch Gynecol Obstet. 2016, PMID: 26969650.
Wang T et al. Robotic-assisted vs. laparoscopic and abdominal myomectomy for treatment of uterine fibroids: a meta-analysis. Minim Invasive Ther Allied Technol. 2018, PMID: 29490530.
Pundir J et al. Robotic-assisted laparoscopic vs abdominal and laparoscopic myomectomy: systematic review and meta-analysis. J Minim Invasive Gynecol. 2013, PMID: 23453764.
4. Methods:
a) The definition of a “small number of patients” (one of the exclusion criteria) is to be provided. For instance, if the included studies by Advincula et al. 2007 (29 AM/29 RALM) and by Flyckt et al. 2016 (28 LM/ 25 RALM), so is the exclusion of several other studies not comrehensible (see comment on the “Results”)
b) The limited number of search terms (only “robotic surgery” plus procedure) could be a reason for several missing publications (for instance, the PubMed search using the combination of “robotic” and “robotic-assisted” produces different results. For instance, "robotic myomectomy"– 56 results, “laparoscopic and robotic-assisted myomectomy” 82 results, "robotic-assisted myomectomy" 18 results)
c) Please provide information whether restrictions regarding “publication date” were applied or no.
5. Results:
a) Several relevant prospective or retrospective comparisons are missing, e.g.
Swenson CW et al. Comparison of robotic and other minimally invasive routes of hysterectomy for benign indications. Am J Obstet Gynecol. 2016 PMCID: PMC5086293.
--> The study included 8313 hysterectomies for benign indications, including 4527 RAL hysterectomies!
Martínez-Maestre MA et al. Total laparoscopic hysterectomy with and without robotic assistance: a prospective controlled study. Surg Innov. 2014, PMID: 23833240.
ÖzbaÅŸlı E, Güngör M. Comparison of perioperative outcomes among robot-assisted, conventional laparoscopic, and abdominal/open myomectomies. J Turk Ger Gynecol Assoc. 2021 PMID: 34634858;
Gobern JM et al. Comparison of robotic, laparoscopic, and abdominal myomectomy in a community hospital. JSLS. 2013, PMID: 23743382
Nezhat C, Lavie O, Hsu S, Watson J, Barnett O, Lemyre M. Robotic-assisted laparoscopic myomectomy compared with standard laparoscopic myomectomy--a retrospective matched control study. Fertil Steril. 2009, PMID: 18377901.
Chen YC et al. Comparison of robotic assisted laparoscopic myomectomy with barbed sutures and traditional laparoscopic myomectomy with barbed sutures. Taiwan J Obstet Gynecol. 2018, PMID: 30342656.
Ranes M et al. Robot-Assisted Laparoscopic Myomectomy Versus Abdominal Myomectomy A Retrospective Comparison of Short-Term Surgical Outcomes. J Reprod Med. 2016, PMID: 30383937.
Göçmen A et al. Comparison of robotic-assisted laparoscopic myomectomy outcomes with laparoscopic myomectomy. Arch Gynecol Obstet. 2013, PMID: 22933121.
Griffin L et al. Postoperative outcomes after robotic versus abdominal myomectomy. JSLS. 2013, PMID: 24018077.
6. Results and Discussion:
The outcomes are provided only in regard to the procedure, but not systematically presented and not systematically discussed with regard to outcomes.
This shortcoming is of critical importance especially in relation to complications (for comparison see the above meta-analyses and: Marra AR et al. Infectious complications of laparoscopic and robotic hysterectomy: a systematic literature review and meta-analysis. Int J Gynecol Cancer. 2019, PMID: 30833440)
I will be glad to re-review the manuscript after necessary improvements, since an updated systematic review on this subject will certainly be welcomed by readers. Good luck!
Round 2
Reviewer 2 Report
The manuscript has improved greatly. I appreciate the authors' efforts and recommend accepting their work for publication.